# Effect of Type of Forest Growth Conditions and Climate Elements on the Dynamics of Radial Growth in English Oak (*Quercus robur* L.) of Early and Late Phenological Forms

Andrey I. Milenin [1], Anna A. Popova [2] and Konstantin A. Shestibratov [3,*]

1   Department of Forestry, Forest Taxation and Forest Management, Voronezh State University of Forestry and Technologies Named after G.F. Morozov, 394087 Voronezh, Russia
2   Department of Botany and Plant Physiology, Voronezh State University of Forestry and Technologies Named after G.F. Morozov, Timiryazeva Str. 8, 394087 Voronezh, Russia
3   Branch of the Shemyakin-Ovchinnikov Institute of Bioorganic Chemistry of the Russian Academy of Sciences, Prospekt Nauki 6, 142290 Pushchino, Russia
*   Correspondence: schestibratov.k@yandex.ru

**Abstract:** The pattern of annual radial growth is influenced by various factors: the local growth conditions, the age structure, and the ecotypes or provenances of trees. A more in-depth approach to the study of specific growth patterns of tree forms is needed to predict the further genesis of forests. This research was carried out on healthy English oak trees of early (EF) and late (LF) phenological forms in Shipov Forest, Voronezh Region. The dendroclimatic analysis was performed on permanent sample plots in wet, dry, and very dry oak stands grown on different soil types. The effect of precipitation on annual ring width was assessed using a one-way ANOVA. The LF showed higher radial growth rates on wet sites than the EF did on dry ones. Their annual radial growth was less stable and more variable compared with the LF. For both phenoforms, the most important radial growth factors are the composite indicators reflecting the ratio of temperature and moisture (Selyaninov's hydrothermal coefficient and Lang's rain factor). Generally, the radial growth minima coincided in time on dry and wet sites, and the periods of maximum growth were associated with high-water years.

**Keywords:** English oak; climatic response; type of local growth conditions; age trend; phenological forms

## 1. Introduction

The English oak (*Quercus robur* L.) is one of the most important forest species in Europe and Russia. Under the influence of various environmental conditions, the English oak has developed numerous intraspecific taxonomic categories with specific bioecological properties: floodplain, upland, and ravine ecotypes; saline cretaceous and boreal edaphotypes; as well as numerous morphological forms that differ in crown, bark, leaves, fruits, types of branching, and other features. According to the International Code of Botanical Nomenclature, a taxonomic rank, or form, is used within a species.

The phenological forms within the species *Quercus robur* L. differ in the timing of leafing. The early (*Quercus robur* L. f. praecox Czern.) and late (*Quercus robur* L. f. tardiflora Czern.) phenological forms were first described by E.F. Zyablovsky in 1804 and V.M. Chernyaev in 1858 [1]. In addition to these extreme forms, medium and winter-leafed forms are also sometimes distinguished. The difference between the two forms in growing degree units (at the base temperature of 5 °C) is more than 150 °C according to [2] and up to 350 °C according to [3]. The difference in the time of leafing onset between the early and late forms ranges from 2 to 4 weeks [4–7]. As shown by studies of the genetic determination of phenological forms in oak, these forms separated due to a mismatch in their flowering

periods [8,9]. Thus, the gene flow between them is limited [5], and their genetic differences are confirmed by molecular studies [10,11].

Studies of the oak phenoforms in similar soil and moisture regimes show no clear relationship between the leafing time and such factors as tree age, size, and position in the stand, which suggests that the phenological variability may have a genetic basis [12]. The available comparative studies of the long-term dynamics of the growth and phenology of leaves, the xylogenesis of the two phenological forms, and their sensitivity to meteorological factors [13–17] show the absence of a homogeneous climatic response and the presence of individual patterns of leaf and xylem development, which indicates a large genetic variability [14].

The phenological forms are interesting primarily because of the difference in responses to growth conditions, extreme climatic factors, and pest outbreaks. Most scientists note that the late-forming trees have a shorter growing season, are not damaged by spring frosts, grow faster, have higher quality wood, and are resistant to oak forest pests, such as powdery mildew (*Erisiphe alphitoides*), phytophthora, Corythucha lace bug arcuata, and different species of gall wasps (g. Cynips) [9,16,18,19].

The crown height of the early form is 15–20% higher than that of the late one [20,21]. In late-forming stands, there are usually more trees with valuable economic traits [21], the so-called "plus trees" [22,23]. Although young trees of the early form grow faster [17,21], the growth energy of the late phenological form increases with age [19,24].

Local growth conditions influence the growth rate of both phenological forms [4,23], but the most decisive effect on leafing is that of air temperatures [2,25]. In addition to the growth energy and leafing time, there are morphological differences as well: the late form often has straight, full-woody trunks and highly raised crowns, while the early form has wide, spreading crowns and curved trunks [9,26]. In humid soils with a threat from frost, the late form of oak has an advantage over the early one. In drier conditions, the early form is less damaged by spring frosts and has an advantage over the late form in terms of growth rate. It is thought to make better use of favorable spring moisture conditions but is more vulnerable to defoliation by late frosts and insects. An earlier start of vegetation allows it to start growing before the onset of summer droughts [6,13,27]. On the other hand, the late form avoids the spring threat of defoliation caused by frost and insects. Its growth is more shifted to the summer due to a later start of its growing season. Hence, it is more vulnerable to summer droughts. Therefore, it occurs on more humid soils [27–29]. The study of the biological and ecological features of phenoforms is important from the silvicultural perspective: a selection of an improper phenological oak form for specific moisture conditions is the cause of failures in forest management and oak stand reduction [18,27,30].

Inter- and intra-population differences in adaptivity are observed even at the climatic boundary of the species' occurrence [31,32]. Some argue that the inter-phenotype differences are more pronounced under optimal rather than unfavorable growth conditions [33,34]. However, we assume that there are differences in the annual ring growth between the phenological forms in the forest-steppe region at the species range boundary. One phenoform may be more adaptive than the other in specific local growth conditions. The aim of the study was to investigate the growth patterns of two phenological forms of the English oak at the boundary of their range under the forest-steppe zone in different types of local growth conditions.

## 2. Materials and Methods

The research was carried out on healthy English oak trees of early and late phenotypes in Shipov Forest, Voronezh Region (lat 50°46′00″ N, long 40°20′00″ E) (Figure 1).

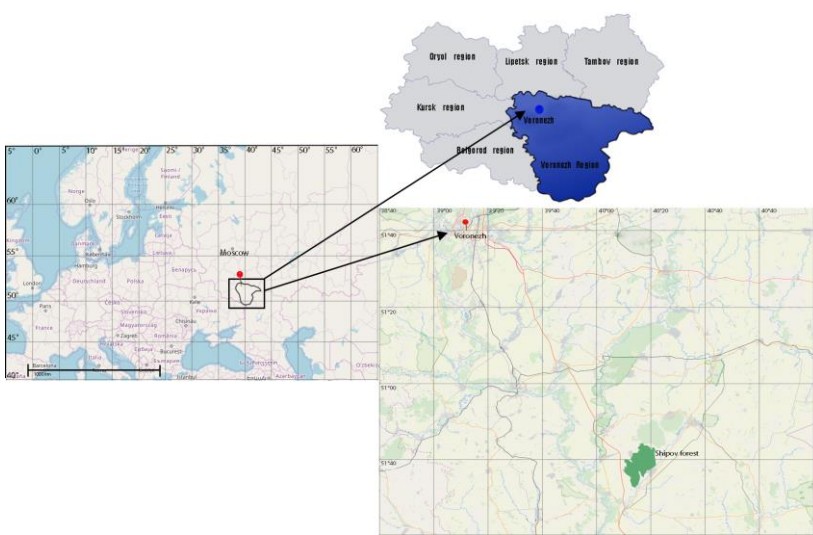

**Figure 1.** Study area: Shipov Forest, Voronezh Region, Central Federal District, Russia.

### 2.1. Edaphic and Orographic Conditions in Shipov Forest

There are several soil types in Shipov Forest, where the research was conducted: haplic chernozems, luvic chernozems, sodic soils, and eutricfluvisols (according to the FAO-UNESCO Soil Map of the World [35]).

The above fragment of a soil map shows the location of the forest area at the intersection of different types of soils. Studies of Shipov Forest by the Russian scientist G.F. Morozov showed great ecological variability associated with the diversity of edapho-orographic conditions. The gentle south- and east-facing slopes have light gray forests and solonetzic soils. The bottoms and lower parts of the gullies and slopes are occupied by alluvial soils of various mechanical compositions. Humus-carbonate soils are formed where chalk outcrops on the surface. Of the above-mentioned soils, gray forest soils are the most common, followed by dark gray forest soils. Gully-alluvial soils are much less common. Solonetzic and humus-carbonate soils are quite rare [36]. Thus, Shipov Forest has unique growing conditions for the English oak, which have led to the emergence of diverse local growth conditions and forest plant communities.

### 2.2. Climatic Profile of the Study Area

According to the bioclimatic classification by Rivas-Martínez et al., 2002 [37], the study area falls into the temperate xeric bioclimate, submediterranean bioclimatic variant.

According to the Krasny Cordon Weather Station (lat 50°66′82″ N, long 40°36′01″ E), the multi-year average temperatures in the study area Shipov Forest are as follows: +17.7 °C in summer, 8 °C in winter, and the average annual temperature is +5.2 °C. The growing period with temperatures above +5 °C is 188 days long. The average last spring and first autumn freeze dates in the study area are 12 April and 29 September. Late spring frosts that affect immature spring shoots of the early phenological form occur nearly every third year. In Shipov Forest, the average annual precipitation is 517 mm. The presented data are similar to those of the cities of Boguchar and Kalach, which are geographically close to Shipov Forest [37].

Although annual precipitation is fairly significant, it is very uneven. In wet years, the amount of precipitation increases, e.g., up to 918 mm fell in Shipov Forest in 1925. In dry years, the amount of precipitation drops, with an absolute minimum of 295 mm. Most precipitation falls in July, followed by August, June, and May. Dry periods with high air temperatures are quite frequent in this area. Dry winds are also typical and most frequent in the second half of April and in May.

According to the Krasny Cordon Weather Station of Shipov Forest, the amount of precipitation has been decreasing over the past 100 years, while the average annual air temperature has increased by nearly 1 °C (Figure 2).

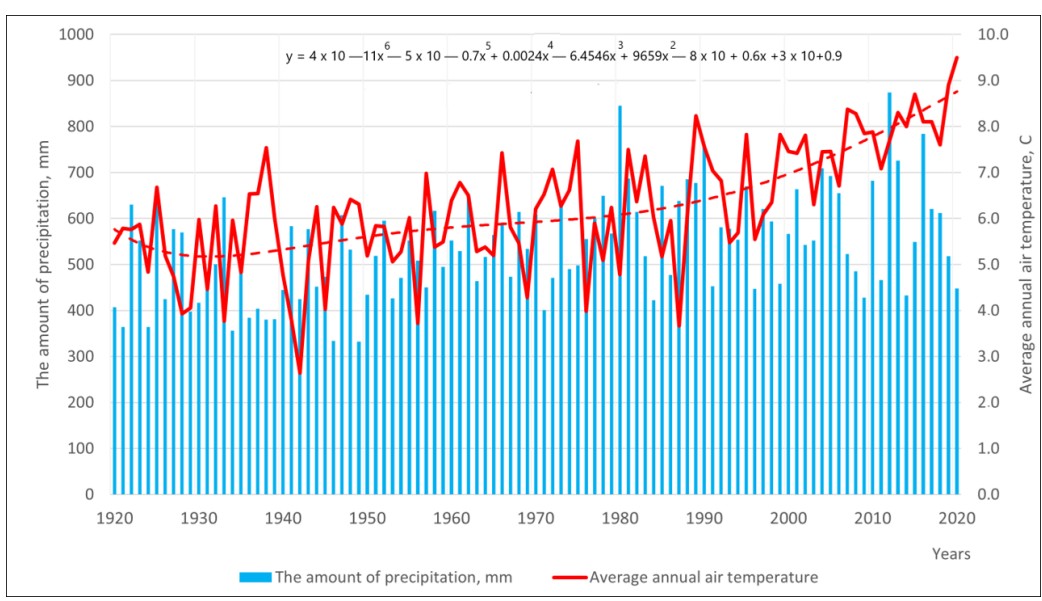

**Figure 2.** Precipitation and average annual air temperatures during the period of 1920–2020, according to the Shipov Forest Weather Station Krasny Cordon.

### 2.3. Sampling Method

Stem cores and discs were obtained on permanent sample plots in the Krasny Forestry (Voronezh Region, Russia), in natural forest stands located at the same heights with similar reliefs. The sample plots are laid out in such a way to allow for the study of trees that grow in places with different soil and moisture regimes that determine the local growth conditions. The sample plots are dominated by *Quercus robur* L. in early and late phenological forms. We took samples from 1400 trees (200 trees per sample plot). The dendroclimatic analysis of the English oak growth (in the upland oakery of Shipov Forest) was carried out on permanent sample plots in wet, dry, as well as very dry oak stands on different types of soils. The local growth conditions of the sample plot were: 1—goutweed oak stand, wet oak grove; 2—goutweed oak forest, wet oak grove; 3—sedge and goutweed oak stand, dry oak grove; 4—sedge and goutweed oak stand, dry oak grove; 5—sedge and grass oak stand, very dry oak grove; 6—sedge and grass oak stand, dry oak grove; 7—grass oak stand, dry oak grove.

The detailed characteristics of the sample plots are presented in Table 1.

**Table 1.** Silvicultural characteristics of the forests.

| Sample Plot No. | Stand Composition | Origin | Mean Age, yrs | Mean Height, m | Mean Stem Diameter, cm | Soil Type |
|---|---|---|---|---|---|---|
| 1 | 1st storey: 90% O 10% A; 2nd storey: 100% M + st P | Natural, by seed | 170 | 33.1 | 54.0 | Degraded chernozem |
| 2 | 1st storey: 80% O 20% A; 2nd storey: 10% M, st L | Natural, by seed | 170 | 31.4 | 49.0 | Degraded chernozem |
| 3 | 90% O 10% A + st M, P | Natural, by coppice | 75 | 21.1 | 24.3 | Light brown, forest, loamy |
| 4 | 50% O 30% A 10% M 10% L | Natural, by coppice | 75 | 22.1 | 26.6 | Light brown, forest, loamy |
| 5 | 100% O st A | Natural, by coppice | 95 | 17.0 | 23.2 | Light gray, solonetzic |
| 6 | 100% O | Natural, by coppice | 95 | 16.0 | 26.0 | Gray, forest, solonetzic |
| 7 | 80% O 10% A 10% L | Natural, by coppice | 78 | 21.8 | 24.8 | Humus-carbonate on cretacious deposits |

Note: O—oaks; A—ash trees; M—maples; L—lindens; P—pear trees; st—single trees.

The phenological form was determined by phenological observations during a route survey (from the Krasny Cordon to the Chernavsky Cordon, the Krasny Forestry) and analysis of samples collected on the study plots. The observations began with the start of the "breaking leaf buds" phenophase. As a control phenophase, the timing of the beginning of the budding of oak leaves was taken. Observations were repeated at a 5-day interval. The average phenophase $\overline{x}$ was calculated by the formula:

$$\overline{x} = \frac{\sum(x \cdot n)}{\sum n} \tag{1}$$

where $x$ is the observation date of registration of leafing; $n$ is the number of individual trees that have entered the phenophase by the last observation timepoint; and $\Sigma n$ is the total number of recorded individual trees in the plot.

The health of the trees was assessed in accordance with the current Rules of Sanitary Safety in the Forests of the Russian Federation, according to which they were categorized as healthy, weakened, severely weakened, wilting, and dead trees and stands in general [38].

### 2.4. Preparation for Macroscopic Measurements

Wood cores and discs for dendrochronological analysis were collected using an increment borer from first-rate average-growth type oak trees growing on a sample plot at a height of 130 cm. After sampling, the collected wood cores and discs were air-dried, fixed to wooden mounts, and sanded. At the beginning, visual cross-dating was carried out, and then the prepared cores were scanned, the tree age was determined, and the annual ring width was measured (to an accuracy of $\pm 0.01$ mm) using the LINTAB 6 ring-width measuring system (Rinntech Company, Heidelberg, Germany); the collected data were analyzed using the TSAP-Win software professional programs.

### 2.5. Statistical Analysis

The average annual ring width (ARW) was calculated based on stem core measurements made with an accuracy of $\pm 5\%$, by Equation (2):

$$a_i = 1/n \sum_{j=1}^{n} x_i \times \gamma \tag{2}$$

where $a_i$ is the average width of annual rings of all sampled trees in an i-th year; n is the number of sampled trees; and $x_{i*}\gamma$ is the annual ring width of a *j-th* tree in an *i-th year*.

The effect of age was excluded by smoothing the source data over an 11-year period using Equation (3):

$$A11i = \sum_{i+5}^{i-5} ai/11 \tag{3}$$

where A11i is the annual ring width smoothed over the 11-year period.

The relative index method is used to take account of factors that depend on individual features of trees or specific conditions at the growth site. The method consists in using a series of relative indices H1; H2; H3; H4 ... Hn, each of which is obtained as using Equation (4):

$$Hi = ai/A11i \tag{4}$$

The thus obtained annual values and the relative indices were plotted (Figure 3).

Shipov Forest is located at the boundary of the forest-steppe and steppe zones; the limiting climate factor is precipitation. To assess the effect of precipitation on radial growth in the English oak, we analyzed the obtained data by one-way ANOVA and calculated the effect size indices [39,40]. The relative indices (Hi) were used as effective features—

total precipitation by year—as gradations of factors. Precipitation data for the period of 1913–2004 were collected at the Shipov Forest Weather Station (Krasny Cordon).

The index of effect size was calculated as a ratio of the factorial sum of squares (Dx) to the total sum of squares dispersion complex (Dy), by Equation (5):

$$\eta x^2 = Dx/Dy, \tag{5}$$

The factorial sum of squares was calculated by Equation (6):

$$Dx = \sum(\sum xi)^2/nai - (\sum x)^2/N, \tag{6}$$

The total sum of squares was calculated by Equation (7):

$$Dy = \sum x^2 - (\sum x)^2/N, \tag{7}$$

Regression analysis was used to identify causal relationships between the meteorological factors and the radial increment. Regression equations were made based on annual ring width measurements, and the changes in radial increment were mathematically analyzed in relation to Selyaninov's hydrothermal coefficient, Lang's rain factor, precipitation, as well as average temperatures over a year, the growing season, and the period of maximum growth. The values of relative indices were used as the resulting feature, and the above-mentioned climate factors were used as independent variables.

The empirical data on the annual radial width and meteorological factors were entered into the Approx software (Russia), and the best fit was calculated by the Fisher test using least squares data fitting.

The absolute values of ARW had an abnormal distribution due to the observed age trend. The ARW indices had a close-to-normal distribution.

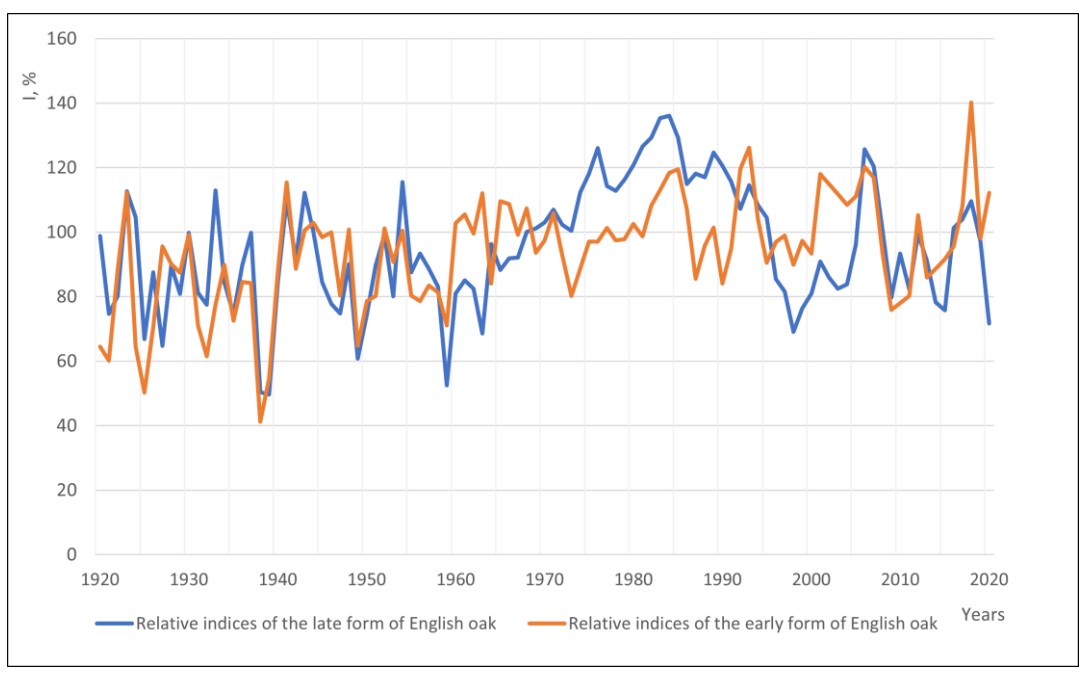

**Figure 3.** The dynamics of the relative indices of early and late wood in the wet goutweed oak stand.

## 3. Results

### 3.1. Annual Ring Width (ARW)

The absolute minimum of ARW in Shipov Forest was recorded in 1939 and was 0.63 mm (relative indices of ARW are shown at Figure 3). The radial growth depression was mainly due to the summer droughts of 1938–1939. According to the Shipov Forest

Weather Station Krasny Cordon, the year of 1938 was especially dry, which set off the decline. In 1938, the precipitation was 387.7 mm, while the average annual amount is 517.9 mm. A decrease in radial growth was noted in 1972 (ARW = 1.05 mm, late wood width = 0.55 mm; relative indices are shown in Figure 3).

The year 1949 was the driest of all years on record in Shipov Forest (295.1 mm of precipitation); that year, there was also a drop in ARW growth.

The dynamics of radial growth show an age-related trend confirmed by decreases in the annual ring width and late wood width (Table 2).

**Table 2.** Statistics on annual ring width and late wood width, by oak age.

| Age, yrs | Annual Width | Statistics | | | | | Late Wood Percentage, % |
| | | Xavg ± m, mm | ±σ, mm | CV,% | *p* | *t*-Test | |
|---|---|---|---|---|---|---|---|
| 10 | ARW | 2.82 ± 0.097 | 0.44 | 15.5 | <0.001 | 3.65 | 75 |
| | EWW | 0.77 ± 0015 | 0.07 | 0.9 | <0.05 | 2.20 | |
| | LWW | 2.05 ± 0.087 | 0.39 | 19.0 | <0.01 | 2.84 | |
| 30 | ARW | 2.35 ± 0.092 | 0.41 | 17.5 | <0.05 | 2.41 | 66 |
| | EWW | 0.79 ± 0019 | 0.09 | 13.5 | <0.01 | 3.13 | |
| | LWW | 1.56 ± 0.073 | 0.32 | 20.8 | <0.01 | 3.01 | |
| 50 | ARW | 1.52 ± 0.036 | 0.16 | 10.6 | <0.001 | 3.24 | 59 |
| | EWW | 0.62 ± 0025 | 0.11 | 21.1 | <0.001 | 3.94 | |
| | LWW | 0.90 ± 0.023 | 0.12 | 13.3 | <0.001 | 4.41 | |
| 70 | ARW | 1.03 ± 0.023 | 0.11 | 10.4 | <0.001 | 4.12 | 56 |
| | EWW | 0.45 ± 0023 | 0.10 | 19.2 | <0.001 | 3.78 | |
| | LWW | 0.58 ± 0.018 | 0.08 | 13.9 | <0.001 | 3.98 | |
| 90 | ARW | 1.14 ± 0.058 | 0.26 | 22.6 | <0.01 | 2.87 | 57 |
| | EWW | 0.49 ± 0027 | 0.12 | 26.3 | <0.001 | 3.38 | |
| | LWW | 0.65 ± 0.044 | 0.19 | 30.4 | <0.01 | 2.96 | |
| 110 | ARW | 1.55 ± 0.070 | 0.31 | 27.5 | <0.05 | 2.47 | 60 |
| | EWW | 0.62 ± 0041 | 0.18 | 34.3 | <0.01 | 2.69 | |
| | LWW | 0.93 ± 0.051 | 0.23 | 24.7 | <0.05 | 2.88 | |
| 130 | ARW | 1.74 ± 0.059 | 0.26 | 15.2 | <0.001 | 3.45 | 59 |
| | EWW | 0.70 ± 0036 | 0.16 | 29.1 | <0.001 | 3.72 | |
| | LWW | 1.04 ± 0.054 | 0.24 | 23.2 | <0.001 | 3.64 | |
| 150 | ARW | 1.29 ± 0.065 | 0.29 | 22.7 | <0.001 | 3.57 | 57 |
| | EWW | 0.55 ± 0029 | 0.11 | 17.1 | - | 1.78 | |
| | LWW | 0.74 ± 0.046 | 0.21 | 28.2 | <0.001 | 3.65 | |
| 170 | ARW | 1.15 ± 0.072 | 0.31 | 27.3 | <0.01 | 2.75 | 53 |
| | EWW | 0.54 ± 0033 | 0.15 | 25.3 | <0.05 | 2.23 | |
| | LWW | 0.61 ± 0.057 | 0.25 | 40.4 | <0.01 | 2.71 | |

Note: ARW—annual ring width, mm; EWW—early wood width, mm; LWW—late wood width, mm; CV—coefficient of variation, %; ±σ—standard error, mm; *p*—statistical significance (shows the level of statistical significance of differences in ARW and LWW between age group trees for a twenty-year period); *t*-test (if tst >2.18—the differences are significant).

The dynamics of radial growth show a cyclic pattern. The most distinct cycles were those of 2–3, 5–7, 11, 22, 40–50, and 100 years (Figure 3). In the analyzed period, decreases in

radial growth were observed in 1872–1876 and 1969–1975, which correspond to a 100-year cycle.

However, the ARW decreased with tree age non-linearly, and fluctuations were observed (Table 2). The amplitude of ARW fluctuations increased significantly following years of unfavorable weather conditions associated with depression of annual radial wood growth. Age specificities were also observed for the variation coefficients of ARW and LWW (Table 2). In 10-year-old trees, the variation coefficients of ARW and LWW were 15.5% and 19%, respectively. The lowest values of this parameter were found in trees aged 50 (10.6% and 13.3% for ARW and LWW, respectively) and 70 years (10.4% and 13.9% for ARW and LWW, respectively). In trees older than 70 years, the coefficients of variation of ARW and LWW gradually increased. In 170-year-old trees, the coefficients of variation of ARW and LWW were 27.3% and 40.4%, respectively. The relationship between the annual ring width and its coefficient of variation was inverse: the average ARW decreased, while its variation generally increased with increasing oak age.

*3.2. The Influence of Climatic Factors on ARW and Late Wood Width in Different Local Growth Conditions*

Data obtained for trees growing in different local growth conditions (D0—very dry, D1—dry, and D2—wet oak stand) show not only the general influence of climatic factors on ARW in the English oak but also some peculiarities of climatic response depending on the soil type and forest type, as well as differences in the rate of post-depression recovery of growth indicators.

Summer droughts and high air temperatures negatively affected the ARW. Mathematically, the relative indices of the ARW—average summer temperature relationship were determined using the Approx software and can be expressed by Equation (8):

$$Y = 0.42663 \times 10^{\wedge}(21.279/X) \tag{8}$$

which shows that relative indices of ARW reduced as the average summer temperature increased.

The relationship between relative indices of ARW and annual precipitation in the growth conditions of D2 is presented in Figure 4.

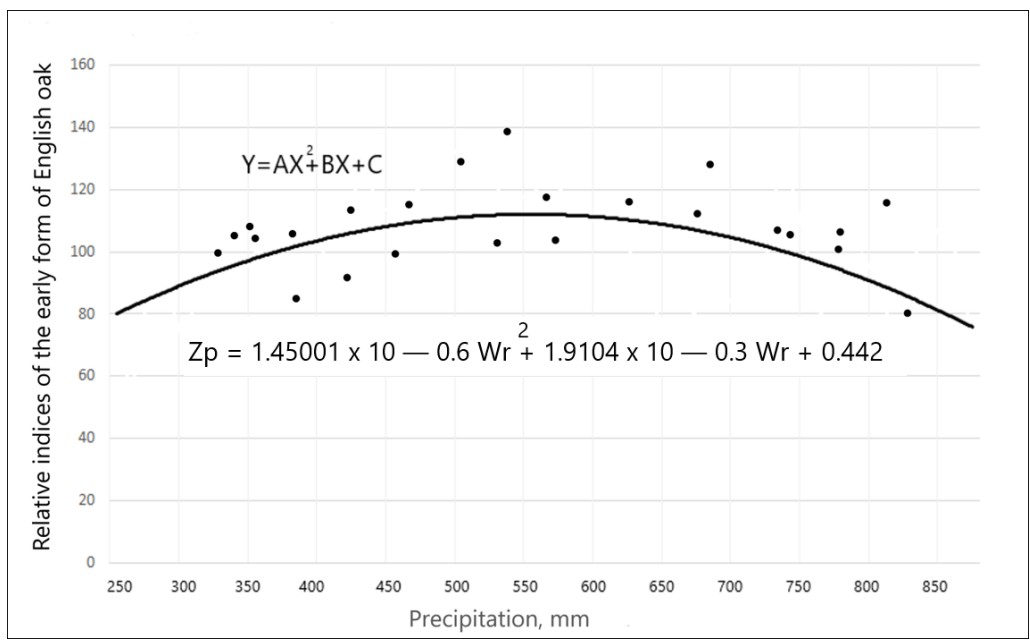

**Figure 4.** Relationship between relative indices of ARW and annual precipitation in the late form of English oak in the wet goutweed oak stand of Shipov Forest.

A decrease in radial growth can be predicted for years with annual precipitation of less than 470 mm or more than 840 mm (Figure 4). An absolute minimum of radial growth in the early phenological form was noted in 1972 (ARW = 0.57 mm, LWW = 0.33 mm). In the same year, there was also a decrease in radial growth in the late forms (ARW = 1.05 mm, LWW = 0.55 mm).

In the dry oak grove D1, oaks of the early phenological form had minimal ARW in the following years: 1921–1925, 1931, 1932–1940, 1943, 1947, 1949–1952, 1954, 1958, 1960, 1968, 1971–1975, 1985–1990, 1995–2002, 2008–2015 (Figures 5 and 6). In those years, the relative indices of ARW varied within 66%–99%.

In late-forming oaks, ARW minima were observed in 1921, 1926, 1931, 1934–1940, 1945–1950, 1954–1958, 1969–1963, 1969–1976, 1982–1986, 1989–1998, 2006–2010, 2012, and 2015, with the relative indices of ARW varying within 55%–99%.

Minimal ARW values generally did not differ between dry and wet oak groves. The absolute minimum in the early oak form was noted in a dry oak grove in 1973, with the ARW index being 66%. In its late form, the absolute minimum was noted in 1984. That year was very dry, with precipitation of 315.7 mm (61% of the long-term annual average) and particularly little precipitation of 9.6 mm during the period of the late, intensive growth in June.

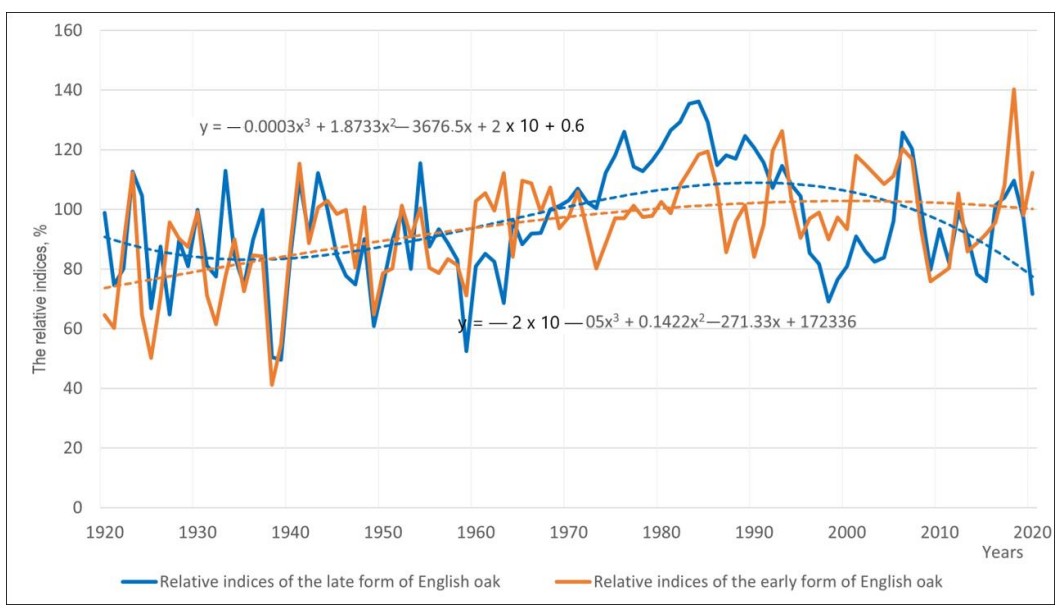

**Figure 5.** Long-term dynamics of relative indices of ARW in English oak in dry oak stand D1.

The periods of maximum ARW in a dry oak grove, as well as in a wet one, are associated with wet years that occasionally follow dry years (Figures 5 and 6). For example, maximum increment can be explained by an aftereffect that manifests itself not only in meteorologically unfavorable but also favorable years. The relative indices of ARW gradually increased after 1973 and peaked in 1982. The values of the annual ring width index and the amount of precipitation were as follows: 1973 – z%—595.1 mm; 1974 –102%—553.1 mm; 1975 –92%—435.4 mm; 1976 –98%—796.4 mm; 1977 –110%—735.8 mm; 1978 –102%—615.3 mm; 1979 –110%—739.7 mm; 1980 –104%—709.1 mm; 1981 –123%—723.4 mm; 1982 —541, 3 mm—142%.

Summing up the results related to the influence of growth conditions and oak form on the long-term dynamics of ARW, one should note that the radial growth in the dry oak stand was less stable than in the wet one. In the conditions of D1, the variation range of the relative indices of ARW in the late form was greater than that in the early one: 55%–142% versus 66%–138%, respectively. In the very dry oak stand on solonetzic soils, the minimal

and maximal radial growths were mostly similar to those in the dry and wet oak stands (Figure 6).

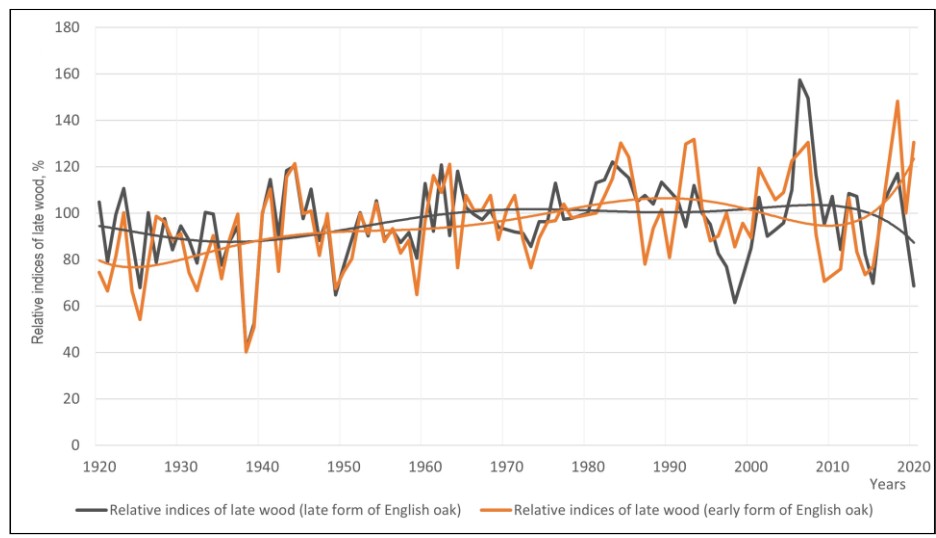

**Figure 6.** Long-term dynamics of ARW in two phenological forms of English oak on solonetzic soils.

In 1924, a very wet year in the period of 1920–1930, the radial growth decreased both in the early and late forms in the wet oak stand D2, but only in the late form in D1 and D0.

The aftereffect was particular strong in the very dry oak stand. Thus, in its early form, the ARW stopped decreasing in the wet, dry, and very dry oak groves after 1972, 1973, and 1974, respectively. The long-term radial growth dynamics showed some peculiarities associated with phenological form and local growth conditions. In a wet oak stand, in young oaks (saplings and polewood), the ARW was greater in the early form; in middle-aged and maturing oaks, it was similar in both forms; and in old oaks, it was greater in the late form. In a dry oak stand, in young oaks (up to 20 years of age), the radial growth was greater in the early-form stand; in older trees, however, it was about the same in both forms. In a very dry oak stand, a greater radial growth was observed in the late form at the initial growth stages and in the early form at the stage of maturity. The ARW stability decreased from optimal (D2) to extreme (D0) growing conditions (the ARW range increased).

The relationship between ARW and summer precipitation is shown in Figure 7.

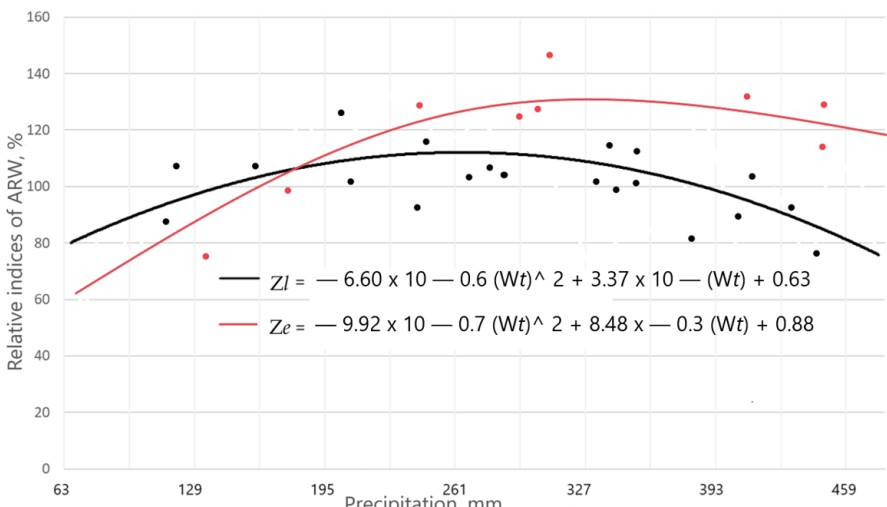

**Figure 7.** Relationship between ARW and summer precipitation in late (Zl) and early (Ze) forms of English oak in a dry sedge oak grove, Shipov Forest.

In the conditions of the dry calciphilous oak stand on humus-carbonate soils, the ARW minima and maxima were noted in the following years: 1941, 1947, 1951, 1957, 1959, 1962, 1968, 1976, 1978, 1979, 1983, 1984, 1985, 1986, and 1988.

Depressions of ARW were observed in 1942, 1954, and 1969, when the precipitation and air temperatures were below the long-term annual averages. Absolute maxima were observed in the years 1950, 1958, and 1964, with above-average annual precipitation and air temperatures.

The long-term dynamics of the oak ARW in a dry calciphilous oak stand is characterized by the presence of two distinct periods of intensive and slow growth corresponding to different age stages. During the first 35 years of growth, the ARW ranged from 2 to 4 mm. In the second period, after the age of 35 years, the ARW decreased abruptly to a range of 1–2 mm. This phenomenon can be explained by the soil factor effect. As is known, humus-carbonate soils are formed on cretaceous sediments and match gray and brown soils in fertility. The root system of a young oak is mainly located in the upper soil horizons, with a weakly alkaline pH close to neutral and increased humus formation. Hence, the growth of the oak is quite intensive. At an older age, the root system reaches the parent rock (dense chalk, sometimes with a lumpy structure), hence the abrupt decrease in growth.

### 3.3. Effects of Climate Factors on ARW

The studied climate factors can be arranged in descending order of their effect on ARW in oak as follows: Selyaninov's hydrothermal coefficient, Lang's rain factor, precipitation, temperature (Table 3). The most important are the complex factors that combine both precipitation and temperature. The effect of these factors increased from the extreme (D0) to optimal growing conditions (D2). The ARW measurements and climate data for the period 1913–1988 were analyzed using one-way ANOVA [41].

**Table 3.** ANOVA results for the effects of climate factors on ARW in English oak.

| Climate Factor | LGCs | Pheno-Form | Effect Size Index | Effect Size Index Error | F-Test | t-Test | |
| --- | --- | --- | --- | --- | --- | --- | --- |
| | | | | | | p = 0.05 | p = 0.01 |
| Annual precipitation | D2 D1 D0 | late | 0.18 | 0.048 | 3.54 | 2.36 2.40 2.36 | 3.3 |
| | | early | 0.06 | 0.050 | 1.20 | | |
| | | late | 0.21 | 0.052 | 4.04 | | |
| | | early | 0.08 | 0.060 | 1.33 | | |
| | | late | 0.13 | 0.050 | 2.60 | | |
| | | early | 0.06 | 0.054 | 1.11 | | |
| Summer precipitation | D2 D1 D0 | late | 0.24 | 0.046 | 5.11 | 2.34 2.40 2.34 | 3.3 |
| | | early | 0.10 | 0.050 | 2.00 | | |
| | | late | 0.30 | 0.030 | 10.0 | | |
| | | early | 0.38 | 0.029 | 13.1 | | |
| | | late | 0.09 | 0.050 | 1.82 | | |
| | | early | 0.29 | 0.040 | 7.25 | | |
| Selyaninov's hydrothermal coefficient | D2 D1 D0 | late | 0.67 | 0.019 | 35.26 | 2.36 2.40 2.36 | 3.3 |
| | | early | 0.71 | 0.016 | 44.37 | | |
| | | late | 0.24 | 0.050 | 4.80 | | |
| | | early | 0.20 | 0.052 | 3.80 | | |
| | | late | 0.34 | 0.038 | 8.95 | | |
| | | early | 0.17 | 0.048 | 3.54 | | |
| Lang's rain factor | D2 D1 D0 | late | 0.20 | 0.046 | 4.35 | 2.35 2.40 2.35 | 3.3 |
| | | early | 0.31 | 0.039 | 7.95 | | |
| | | late | 0.18 | 0.038 | 4.74 | | |
| | | early | 0.15 | 0.040 | 3.75 | | |
| | | late | 0.22 | 0.044 | 5.00 | | |
| | | early | 0.13 | 0.050 | 2.6 | | |

*Composite indicators* span the Selyaninov's hydrothermal coefficient and Lang's rain factor rows.

Note: LGCs—local growth conditions; D0—very dry; D1—dry; D2—wet oak stand; F-test—Fisher's test; *t*-test—Student's test.

The effects of the factors increased from the extreme local growth conditions of a very dry oak stand (D0) to the optimal growth conditions of a wet oak stand (D2). In a wet oak stand, the size of the effect of the hydrothermal coefficient in the early phenological form was greater than in the late one: $\eta_x^2 = 0.71$ and Ff = 44. 37 versus $\eta_x^2 = 0.67$ and Ff = 35.26, respectively. In a dry and very dry oak stand, the index of the hydrothermal coefficient effect size in the late form (D1: $\eta_x^2 = 0.24$; Ff = 4.80; D0: $\eta_x^2 = 0.34$; Ff = 8.95) was higher than in the early form (D1: $\eta_x^2 = 0.20$; Ff = 3.80; D0: $\eta_x^2 = 0.17$; Ff = 3.54).

Precipitation-related climate factors, particularly summer precipitation, rank second in terms of their impact on ARW. The largest effect of this factor was observed in the early-formING oaks in the dry oak stand conditions: $\eta_x^2 = 0.38$.

*3.4. Differences between the Phenological Forms of English Oak Grown in Different Local Growth Conditions*

In the conditions of the upland oak stand in Shipov Forest, the late phenological form has higher growth rates in wet sites than the early form does in dry ones.

The radial growth of the oak is affected by climate factors. They can be arranged as follows in the descending order of their effect: hydrothermal coefficient; rain factor; summer, autumn-winter, and annual precipitation; air temperature during the growing season, summer, and annual average (Table 3). The effect of the climate factors on oak ARW increases from extreme to optimal growth conditions. Stands of the late phenological form are more sensitive to changes in the climate elements.

Summer precipitation is more important in the late form; while autumn and winter precipitation are more important in the early form. The type of relationship between radial growth and climate elements varies with growth-site conditions. In wet sites, growth is depressed by both excessive and deficient precipitation; in very dry ones, only by deficient precipitation. A larger amount of autumn-winter precipitation contributes to increased growth; the effect of this factor increases from very dry to wet soils. An inverse relationship has been established between air temperature and growth: higher temperatures contribute to decreased radial growth, regardless of the phenological form of oaks.

Analysis of ecological factors' influence on the relative indices of ARW has shown the maximal effect of orographic conditions ($\eta_x^2 = 91.8$; Ff = 53.8), soil moisture ($\eta_x^2 = 81.9$; Ff = 17.4) and soil type ($\eta_x^2 = 54.2$; Ff = 7.1).

The stability of radial growth is associated with the stability of phytocenoses. The early phenological forms of oak are less resistant. As a result, their annual radial growth was less stable and varied more than in the late form.

Radial growth stability can be used as an indicator of forest health.

**4. Discussion**

Our study has shown a high level of variability in ARW relative indices and the effects of meteorological factors. Results were obtained for stands at the species range boundary, in the forest-steppe zone of the Central Russian upland. The variability increases with differences in tree age, local growth conditions (including soil type), and phenological forms.

Our data on the effect of tree age on growth is consistent with previously published data for oak [42] and other tree species [43,44]. The highest ARW values are typical for the first decades of tree growth. Then there is a non-linear decrease in growth. Analysis of the ARW time series of the English oak showed a cyclical pattern with 2–3, 5–7, 11, 22, 40–50, and 100-year cycles. The short cycles of tree growth are associated with fluctuations in weather and solar activity [42–47].

The intensive growth during the first decades is associated with the biological features of the plant ontogenesis, and the following deceleration is explained by the gradually increasing competition for resources due to closing crowns [43,48] and developing root systems. The combinations of climatic, orographic, soil, and moisture regimes determine the growth conditions of forest stands. The results of our study show that the age-related

dynamics of annual ring growth are closely related to soil characteristics. The most active and stable growth is observed on chernozem soils in wet oak forests. An oak study in the dry steppe [47] reported the same growth dynamics, but with tree health deteriorating as growth slowed with age.

We have observed that growth minima in the dry oak forests generally coincide in time with those in the wet forests. The periods of maximum growth in dry oak stands, as well as in wet oak stands, were associated with high-water years, which periodically followed dry years. The ARW minima and maxima did not occur in the same years for both phenoforms. In the early phenoform, a decrease in growth was noted in the year of exposure to an adverse factor, and the aftereffect persisted for several years.

Frequent spring defoliation of the early form leads to less stable radial growth, whereas the late phenoform is more resistant to winter and early-spring temperatures [14,17,29].

Studies of ARW patterns in the early and late forms grown on different soils show high variability and nonlinear changes in the growth of late and early wood. Many authors note that different phenoforms often prefer different soil conditions, and the wrong choice of a phenoform for a forest plantation may later cause its degradation [6,14,27,28,49,50]. The phenological dynamics are taken into account when selecting planting material for some regions of Europe and Russia [23,51]. The phenoforms differ in ecological preferences [50], and the late form surpasses the early form in growth intensity with age on all types of soils [3,21].

Our study also showed the differences between the early and late forms of English oak, which are manifested in their occurrence in the local growth conditions of the upland oak grove. In optimal growth conditions, the two phenoforms grow together, with a slight prevalence of the late form. In extreme growth conditions—on humus-carbonate soils and solonetzic—the late form largely prevails. We also found that young oaks of the early phenological form grew faster than those of the late form in both wet and dry oak forests. According to our data, the late phenological form prefers the conditions of very dry oak forests. This finding is confirmed by some studies [9] but contradicts others [52,53]. The most contradictory data relate to the distribution of oak phenoforms depending on soil and local topography [27]. In different parts of the oak distribution area, these patterns may differ. In habitats with smooth relief, there is often a mixture of the phenoforms [54]. With similar slope direction and steepness and similar soil and moisture regimes, the prevalence of a specific oak phenoform is determined by the spring thermal regime of the habitat [55]. Some researchers associate the adaptability of a particular form with its occurrence. For example, in southern Belarus, the late form has a wider adaptability to the habitat [56], which is shown by its greater occurrence. However, the high occurrence of a phenoform may be due to its strong inheritance and acorn concentration around the parent tree. A high level of inheritance was noted for ARW indices [57], wood anatomy [58], and, hence, water use efficiency [59].

In general, a comparison of the results obtained for research objects located in different growth conditions shows the oak's sensitivity to soil moisture. The aftereffect of "climatic stress" is strongly manifested in the long-term dynamics of ARW: in the conditions of a dry oak grove, the growth is less stable than in wet oak grove, which is consistent with the studies of the sensitivity of different oak species (*Quercus pubescens, Quercus frainetto*) to soil moisture conditions [60], and the existing models of moisture sensitivity of oaks in arid areas [61]. Summer droughts and high air temperatures have the greatest impact on the growth, as confirmed by an earlier study of the English oak in the Voronezh region [62]. Similar results were shown for oak (*Quercus petraea, Q. frainetto,* and *Q. cerris*) [63,64], and a systematically similar species of *Fagus sylvatica* [65].

We found differences in the effects of meteorological factors on the phenological forms: most important are composite indicators reflecting the ratio of temperature and moisture. The magnitude of their effect on growth in the phenological forms depends on the type of growth conditions, which is consistent with other studies [66]. The second most important factor is precipitation-related weather elements, primarily summer precipitation. In dry

oak forests, the greatest effect of summer precipitation was observed in its early forms. In wet oak forests, it was greater in the late form. In the very dry oak grove, the effect of summer precipitation in the early-form stand was greater than in the late-form stand. The observed patterns can be explained by the ability of the early form to use abundant soil water before the leafing of the late form, differences in root system structure [52,53], and the timing of vessel formation in early wood [15,54].

## 5. Conclusions

Our findings revealed that ARW varies significantly and that the effects of climatic factors differ significantly between two phenological forms of English oak in the forest-steppe area. This proves our hypothesis about the presence of significant differences between the phenoforms in unfavourable conditions at the boundary of the species' distribution area. The phenological forms have specific ecological needs, which may differ with tree origin, age, health, and local growth conditions (relief, soil, groundwater level, surrounding vegetation, etc.).

The conducted study allows for the following conclusions: the growth and health of forests based on the two studied phenological forms of the English oak depend on the growth conditions. The radial growth of oak is significantly affected by climatic factors. According to their magnitude, they can be arranged in the following descending order: Selyaninov's hydrothermal coefficient; Lang's rain factor; summer, autumn, and winter, and annual precipitation; average air temperatures during the growing season, summer, and annual temperatures. The late phenoform of oak was shown to be more sensitive to changes in climatic factors than the early one.

Summer precipitation is more important for the growth of the late form, whereas autumn and winter precipitation are more important for the early form. Growth conditions can influence radial growth-climatic factor relationships. In wet sites, both low and high amounts of precipitation inhibit the growth, whereas in very dry sites, only low precipitation does. The greater the average autumn and winter precipitation, the greater the radial growth; this factor had the greatest effect in wet growth conditions. We found that air temperatures and radial growth were inversely related, regardless of the phenological form.

Early-form oak forests showed large differences between the minimum and maximum values of the annual ring index. The tree growth stability of the early form is lower than that of the late form. Tree growth stability can be used as a diagnostic marker of forest stand health.

In the upland oak grove of Shipov forest, the late form showed greater radial growth on wet sites than the early form did on dry sites. On wet sites, the growth of the late form was more stable than that of the early form. Generally, the radial growth minima coincided in time on dry and wet sites, and the periods of maximum growth were associated with high-water years. In years with favorable weather conditions, the growth increased equally in both the early and late forms; in adverse years, it fell more significantly in the early form. Radial growth maxima and minima can be explained by the aftereffect of weather conditions, favourable or adverse, respectively. The aftereffect was most pronounced in very dry oak groves. In a dry calciphilous oak forest, one can clearly distinguish two periods of radial growth, intensive and slow, in different age groups and in both phenological forms.

For both phenoforms, the most important radial growth factors are the composite indicators reflecting the ratio of temperature and moisture (Selyaninov's hydrothermal coefficient and Lang's rain factor), reaching their utmost effect in wet growth conditions.

The second most influential weather factors in terms of oak radial growth are precipitation-related ones. The relationship between radial growth and annual precipitation can be described by a second-order parabola. Summer precipitation is of greatest importance for radial growth and peaks in the early growth stands in a dry oak forest. The radial growth of the late form is most influenced by annual precipitation. The effect of the

factor depends on growth conditions and reaches a maximum in the late-forming stands in a dry oak forest. The early phenoform's radial growth is most affected by average autumn and winter precipitation and average annual air temperature. Radial growth is inversely related to air temperature. The effect of the factor increases as the growth conditions worsen.

**Author Contributions:** Conceptualization, A.A.P. and K.A.S.; methodology, A.I.M.; software, A.I.M.; validation, A.I.M. and A.A.P.; formal analysis, A.I.M.; investigation, A.I.M.; data curation, A.I.M.; writing—original draft preparation, A.I.M.; writing—review and editing, A.I.M.; visualization, A.A.P.; supervision, K.A.S.; project administration, K.A.S.; funding acquisition, K.A.S. All authors have read and agreed to the published version of the manuscript.

**Funding:** This work was financially supported by the Russian Science Foundation (Project No. 22-64-00036).

**Data Availability Statement:** Not applicable.

**Acknowledgments:** The authors of the article express their gratitude and special thanks to S.M. Matveev for careful reading of the article and recommendations for the correct interpretation of research data.

**Conflicts of Interest:** The authors declare no conflict of interest.

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
