# Peer review of "Effect of Type of Forest Growth Conditions and Climate Elements on the Dynamics of Radial Growth in English Oak (Quercus robur L.) of Early and Late Phenological Forms"

_forests, doi:10.3390/f14010011_

Round 1

Reviewer 1 Report

The submitted manuscript entitled 'Effect of type of forest growth conditions, climate elements on the dynamics of radial growth in English oak (Quercus robur L.) of early and late phenological varieties', presents the results of dendrochronological research on two morphologically similar but differ in ecology varieties/forms of one of the most important European forest tree species. Chernajev described them in the mid-nineteenth century from Ukraine (Steven 1957). Their occurrence is known only from eastern Europe, mainly from places with a stronger influence of continental climate (Jones 1959, Karazac 1898). In many regions of Europe, foresters and botanists have paid attention to the differences between phenological varieties of oak. Differences in leaf development between early and late varieties reach several weeks (e.g. Puchałka 2017, Slepykh 2016, Wesołowski and Rowinski 2008). Early oak naturally occurs on drier soils. It is believed to make better use of spring's favourable moisture conditions but is more vulnerable to defoliation by late frosts and insects. Earlier vegetation onset allows it to realize growth before the onset of summer droughts (Karazac 1898, Puchałka 2016, 2017, Wesołowski and Rowiński 2008, Rubtsov and Utkina 2008). On the other hand, the late variety avoids the spring threat of defoliation caused by frost and insects. In contrast, the later onset of vegetation causes its growth to shift more to summer. Therefore, it is more vulnerable to summer droughts. Hence, it occurs on wetter soils (Izbdebski 1956, Karazac 1989, Rubtsov and Utkina 2008, Vikhrov 1954). 

Some authors suggest that the improper selection of oak phenological varieties for moisture conditions is the cause of failures in forest management and the decline of oak stands (e.g. Batos et al. 2014; Rubtsov and Utkina 2008). Comparative studies on the long-term growth dynamics and leaf phenology, and xylogenesis of the two oak phenological varieties and their sensitivity to meteorological factors have been published at least several times (e.g., Puchałka 2016, 2017, Kitin 1992, Kostic et al 2021, Koval and Kostyashkin 2015). I believe that the main shortcoming of this manuscript is the lack of scientific hypotheses that are well supported by previous research on oak phenological forms. I think that what could be expected could be based on the studies mentioned above and many other studies of which it is impossible to note them all here. I think it is necessary to expand on the introduction by showing that many studies have been conducted. There are many reproducible patterns from various places in Europe regarding habitat preferences, but the patterns of tree growth dynamics of both varieties are not entirely clear. 

In addition, it lacks an explanation of the subject of the study. A non-Eastern European reader will not know what these oak lesions are and how they differ. It is worth showing here that there is limited gene flow between them because the flowering dates do not overlap (Slepykh 2016). Molecular studies also confirm this (Pirko et al. 2018). It is worthwhile to develop the Introduction, which currently takes a very shallow approach to the topic by ignoring existing knowledge, resulting in a lack of research hypotheses. This should also help develop the Discussion, which is not a strong point of this manuscript. This is also the result of a lack of hypotheses and little insight into the literature related to oak phenological forms. I recommend the publication of Russian authors who summarize information on many comparative studies on early and late oak varieties (Rubtsov and Utkina 2008, Utkina and Rubtsov 2018). Many valuable studies were cited there, which would help to find source literature data. 

The authors can significantly improve the Introduction and Discussion by considering the above comments. The Discussion needs to focus more on the results of comparing the two varieties of oak. The authors should discuss them with other studies on oak phenological varieties and knowledge of their sensitivity to environmental conditions. The cited papers are not strongly related to this. This is also important for Conclusions, which currently look more like a summary of more interesting research results that do not have a hypothesis.

In Materials and methods, an explanation is needed of how the authors identified individuals of the early and late varieties. This is rather impossible in the growing season due to the lack of clear morphological differences. They can be distinguished easily only in spring by the timing of leaf development and in winter, when often on the late variety, the leaves do not fall until winter. What about intermediate individuals? They are usually also present in populations.

Regarding the information in L.99-100. was the effect of late frost damage noticeable in annual growth? Puchalka et al (2016), found no response in increment widths, size and number of vessels after late spring frost in 2007 and 2011. This may be due to the unclear relationship between the timing of leaf development and the onset of the xylogenesis (Puchałka et al. 2017, Sass-Klaassen et al 2011).

L109-114: Materials and methods is the wrong place for summaries and conclusions.

L134: How many trees were sampled? Usually, two cores per tree are taken. Ten trees?

L434-435: Another cause of defoliation may be late spring frost or larvae of Lepidoptera (Karazac 1898, Puchałka et al 2016, Wesołowski and Rowiński 2008, Rubtsov and Utkina 2008). Earlier, the authors wrote that an early variety at the study site had repeated frost damage to young shoots. According to my observations, flowers are also sometimes damaged then. Hence, despite the stress, the plants have no effort to produce acorns. In contrast, acorns are produced in the late variety, which has dormant or swollen leaf buds at this time. It would be worthwhile to include this in the discussion. There are published studies on the effect of seed years and years without seed production on the width of annual growth.

L447-450: Please discuss it with other studies (e.g. Kostic et al 2021) and general knowledge of habitat preferences and ecology of both phenological varieties.

The Conclusions contain unnecessary sentences, e.g.

L.457: This sentence is not needed.

L.558-459: This sentence says nothing new. Too obvious to post in Conclusions.

L465-466: Authors wrote "The effect of the climate 465 factors on oak radial growth increases from extreme to optimal growth conditions." - This was also shown by other studies in provenance tests on spruce (Klisz et al. 2019).

L466-468: Repeated information from the Discussion.

L469-470: Probably because its vegetation is shifted more to summer when there is a water shortage.

L475-476: which variety?

L477-478: Unclear.

The authors must rewrite the Conclusions. With such a good sampling of both phenological varieties, they have an incredible opportunity to verify their knowledge on the differences in susceptibility to meteorological and moisture conditions of oak phenological varieties. However, as I wrote earlier, this requires a deeper look at already published studies. However, as I wrote above, a good Introduction with a research hypothesis is essential to write a strong Discussion and rational conclusions. This may be a very good article, but it will take a lot more effort to improve it.

Kind regards

References

Batos B, Šešlija Jovanović D, Miljković D (2014) Spatial and temporal variability of flowering in the pedunculate oak (Quercus robur L.). Šumarski List 7–8:371–379

Izdebski K (1956) Wstępne badania nad ekologią i rozmieszczeniem dębu szypułkowego (Quercus robur L .) w Polsce. Ann UMCS, C 11:415–506 (in Polish with English summary)

Jones EW (1959) Biological Flora of the British Isles. Quercus L. J Ecol 47:169–222

Kitin P (1992) Dynamics of cambial activity in the stem of early- and late-flushing forms of oak (Quercus robur vars. praecox and tardiflora) in the Park of Freedom, Sofia. Nauk za Gorata 27:3–13

Klisz M, Buras A, Sass-Klaassen U, et al (2019) Limitations at the Limit? Diminishing of Genetic Effects in Norway Spruce Provenance Trials. Front Plant Sci 10:306. https://doi.org/10.3389/fpls.2019.00306

Kozarac J (1898) Kasni (pozni) hrast (Quercus pedunculata var. tardissima Simonkai). Sumar List 22:41–53

Kostic S, Orlovic S, Karaklic V, et al (2021) Allometry and Post-Drought Growth Resilience of Pedunculate Oak ( Quercus robur L .) Varieties. Forests 12:930. https://doi.org/10.3390/f12070930

Koval IM, Kostyashkin DC (2015) The influence of climate and recreation on formation of layers of annual wood of early and late forms Quercus robur L. in Kharkiv Greenbelt. Sci Bull UNFU 25:52–58

Pirko YV, Netsvetov M, Kalafat LO, et al (2018) Genetic features of the phenological forms of Quercus robur (Fagaceae) according to the analysis of the introns polymorphism of β-tubulin genes and microsatellite loci. Ukr Bot J 75:489–500. https://doi.org/https://doi.org/10.15407/ukrbotj75.05.489 

Puchałka R, Koprowski M, Przybylak J, et al (2016) Did the late spring frost in 2007 and 2011 affect tree-ring width and earlywood vessel size in Pedunculate oak (Quercus robur) in northern Poland? Int J Biometeorol 60:1143–1150. https://doi.org/10.1007/s00484-015-1107-6

Puchałka R, Koprowski M, Gričar J, Przybylak R (2017) Does tree-ring formation follow leaf phenology in Pedunculate oak (Quercus robur L.)? Eur J For Res 136:259–268. https://doi.org/10.1007/s10342-017-1026-7

Rubtsov V V, Utkina IA (2008) Adaptatsionnyye reaktsii duba na defoliatsiyu. M.: Grif i K, Moskva

Sass-Klaassen U, Sabajo CR, den Ouden J (2011) Vessel formation in relation to leaf phenology in Pedunculate oak and European ash. Dendrochronologia 29:171–175. doi:10.1016/j.dendro.2011.01. 002

Slepykh OO (2016) Rhythm of phenology and distribution phenological forms of Pedunculate oak (Quercus robur L.) in Donetsk region. Biol Syst 8:272–279

Steven C V (1857) Verzeichnis der auf taurischen Halbinsel wildwachseden Pflanzen. Bull la Société Imp des Nat Moscou 30:325–398

Šafar J (1966) Problem fizioloških, ekoloških i ekonomskih karakteristika kasnoga i ranog hrasta lužnjaka. Sumar List 11–12:503–515

Utkina IA, Rubtsov V V. (2018) Studies of Phenological Forms of Pedunculate Oak. Contemp Probl Ecol 10:804–811. https://doi.org/10.1134/s1995425517070101

Vikhrov VE (1954) Stroenie i fiziko-mekhanicheskie svoistva drevesiny duba. Izd-vo Akademii nauk SSSR, Moskva

Wesołowski T, Rowiński P (2008) Late leaf development in pedunculate oak (Quercus robur): An antiherbivore defence? Scand J For Res 23:386–394. https://doi.org/10.1080/02827580802419026

Author Response

Dear Reviewer, we thank you for your work on our manuscript and for your comments, references on the subject. We understood the need to rework the text, it was necessary to deepen the introduction and discussion, therefore these sections of the manuscript are the most modified. All changed parts of the text are highlighted in blue.

Reviewer #1

1) I think it is necessary to expand on the introduction by showing that many studies have been conducted. There are many reproducible patterns from various places in Europe regarding habitat preferences, but the patterns of tree growth dynamics of both varieties are not entirely clear. 

In addition, it lacks an explanation of the subject of the study. A non-Eastern European reader will not know what these oak lesions are and how they differ. It is worth showing here that there is limited gene flow between them because the flowering dates do not overlap (Slepykh 2016). Molecular studies also confirm this (Pirko et al. 2018). It is worthwhile to develop the Introduction, which currently takes a very shallow approach to the topic by ignoring existing knowledge, resulting in a lack of research hypotheses.

Reply: The Introduction has been completely rewritten taking into account your recommendations.

2) The Discussion needs to focus more on the results of comparing the two varieties of oak. The authors should discuss them with other studies on oak phenological varieties and knowledge of their sensitivity to environmental conditions. The cited papers are not strongly related to this. This is also important for Conclusions, which currently look more like a summary of more interesting research results that do not have a hypothesis.

Reply: The Discussion has been completely rewritten taking into account your recommendations.

1) In Materials and methods, an explanation is needed of how the authors identified individuals of the early and late varieties. This is rather impossible in the growing season due to the lack of clear morphological differences. They can be distinguished easily only in spring by the timing of leaf development and in winter, when often on the late variety, the leaves do not fall until winter. What about intermediate individuals? They are usually also present in populations.

Reply: The following information has been added to section 2.3:

“Phenological forms were established by phenological observations in the studied forest types during a route survey (the route ran from the Osered River floodplain at the Red Cordon (lat 50.664906 N, log 40.363079 E) to the Chernavsky cordon, the Red Forestry) and during the analysis of sample plots. The accounting of trees began with their entry into the phenophase "bud opening". As a control phenophase, the timing of the beginning of the budding of oak leaves was taken. Observations were carried out at intervals of 5 days. The calculation of the average phenophase (x ) was established using equation 1:       

x ̅=(∑(xn))/Σn                          (1),

where х - observation dates given by the month of registration of this phenophase; n - the number of individuals who entered the phenophase by the last observation;   Σn - the total number of registered individuals in the plot.

The health of the tree assessed in accordance with the current rules of sanitary safety in the forests of the Russian Federation, according to which they are allocated to healthy, weakened, severely weakened, shrinking and dead trees and plantings in general [38].

The intermediate forms are found, but they were not at the sites studied

2) Regarding the information in L.99-100. was the effect of late frost damage noticeable in annual growth? Puchalka et al (2016), found no response in increment widths, size and number of vessels after late spring frost in 2007 and 2011. This may be due to the unclear relationship between the timing of leaf development and the onset of the xylogenesis (Puchałka et al. 2017, Sass-Klaassen et al 2011).

Reply: The effect of late frosts on annual growth was noticeable, but we did not conduct detailed studies on this issue (updated data on frosts are needed).

3) L109-114: Materials and methods is the wrong place for summaries and conclusions.

Reply: This part of the text has been deleted

4) L134: How many trees were sampled? Usually, two cores per tree are taken. Ten trees?

Reply: We took a total samples of 1400 trees (200 trees per sample plot).

5) L434-435: Another cause of defoliation may be late spring frost or larvae of Lepidoptera (Karazac 1898, Puchałka et al 2016, Wesołowski and Rowiński 2008, Rubtsov and Utkina 2008). Earlier, the authors wrote that an early variety at the study site had repeated frost damage to young shoots. According to my observations, flowers are also sometimes damaged then. Hence, despite the stress, the plants have no effort to produce acorns. In contrast, acorns are produced in the late variety, which has dormant or swollen leaf buds at this time. It would be worthwhile to include this in the discussion. There are published studies on the effect of seed years and years without seed production on the width of annual growth.

Reply: To discuss this issue, it is necessary to compare the data on the width of the annual ring with the harvest years, in early and late oak, they do not always coincide, additional collection of information is needed, which can be solved in further research.

6) L447-450: Please discuss it with other studies (e.g. Kostic et al 2021) and general knowledge of habitat preferences and ecology of both phenological varieties.

Reply: The discussion has been completely rewritten taking into account your recommendations.

7) The Conclusions contain unnecessary sentences, e.g.

L.457: This sentence is not needed.

L.558-459: This sentence says nothing new. Too obvious to post in Conclusions.

L465-466: Authors wrote "The effect of the climate 465 factors on oak radial growth increases from extreme to optimal growth conditions." - This was also shown by other studies in provenance tests on spruce (Klisz et al. 2019).

L466-468: Repeated information from the Discussion.

L469-470: Probably because its vegetation is shifted more to summer when there is a water shortage.

L475-476: which variety?

L477-478: Unclear.

The authors must rewrite the Conclusions. With such a good sampling of both phenological varieties, they have an incredible opportunity to verify their knowledge on the differences in susceptibility to meteorological and moisture conditions of oak phenological varieties. However, as I wrote earlier, this requires a deeper look at already published studies. However, as I wrote above, a good Introduction with a research hypothesis is essential to write a strong Discussion and rational conclusions. This may be a very good article, but it will take a lot more effort to improve it.

Reply: The Conclusion has been completely rewritten taking into account your recommendations.

Reviewer 2 Report

The article has many strengths, but it is important to mention the weaknesses, which should be carefully examined by the authors and appropriate modifications made. 

1. The early and late phenological groups, which are common throughout the range of Quercus robur, are not regarded as taxa, and in my opinion, it would be better to call them 'races' rather than 'varieties'. Races are groups of individuals with no taxonomic rank, defined by a particular character or set of characters. I would therefore suggest considering whether it would not be more accurate to use this term consistently throughout the article.

2. I suggest that the abstract should be supplemented by the most important finding of the study. 

3. The Latin names of the taxa should be in italics throughout the text. 

4. I think that the aim of the paper is not properly formulated. The results of the study should explain the phenomenon, but not simple differences in growth within the one forest. A hypothesis should also be formulated. It is possible to formulate the hypothesis that oaks of a certain race are more adaptive than oaks of another race. 

5. How were the races determined in the field and how was the health (line 62) of the tree assessed? These are very important methodological considerations that need to be described. 

6. I think that the term 'storey' is completely inappropriate for soil (Table 1). Soil is made up of layers.

7. Line 148. The sentence is unclear and needs to be clarified.

8. The methodology is not sufficiently clear, especially the statistical part. It is written that ANOVA was used, but it is not stated which one (one-way, two-way, nested)? How were the results of the radial increments assessed? Whether the data sets were normally distributed or non-normally distributed? Only by knowing the answers to these questions is it possible to assess the interpretation of the results.

9. There are unexplained abbreviations in the tables (e.g., Table 3).

10. What is indicated next to the mean? Standard error or standard deviation?

11. How should the statements in lines 206 to 212 onwards, which deal with tree growth in 1938, 1939, etc., be understood, when all the graphs and tables indicate that growth has been estimated since 1958? And then how to relate the information given in Figure 6, Figure 10, etc. If different periods were taken, then it is necessary to explain all the reasons and provisions in the methodology.

12. As different graphs and tables cover different time periods; the text of the results is difficult to analyse. I would recommend that the authors not only supplement the methodology but also consistently revise the text of the results so that the references to the figures do not contradict the information they contain (e.g., Figure 10 and 11). 

13. Line 388. What are those specific features?

14. Line 420. What is P. sylvestris and what is Q. humilis? The rules of nomenclature must be followed, which state that the genus name may only be abbreviated when the abbreviation does not cause ambiguity. 

15. I suggest reorganising the Conclusions section. Some of the statements should be moved to the discussion section and the sources of information (on race resistance) should be indicated. 

16. The authors should edit the content of the paper to avoid unexplained abbreviations (Table 5), contradictions and inaccuracies. 

Author Response

Dear Reviewer, we thank you for your work on our manuscript and for your comments. The comments of another reviewer are of a analogous type, it was necessary to deepen the introduction and discussion, therefore these sections of the manuscript are the most modified. All changed parts of the text are highlighted in blue.

Reviewer #2

1) The early and late phenological groups, which are common throughout the range of Quercus robur, are not regarded as taxa, and in my opinion, it would be better to call them 'races' rather than ''. Races are groups of individuals with no taxonomic rank, defined by a particular character or set of characters. I would therefore suggest considering whether it would not be more accurate to use this term consistently throughout the article.

Reply: We thank the respected reviewer for clarifying the terminology, we agree with the opinion that the use of the term "varieties" is failed. After analyzing a number of articles, we assume that a more generally accepted variant of the naming of two phenological groups is the term "forms". In the article, the term "varieties" is changed to "forms".

  1. Slepykh, O.O. Rhythm of phenology and distribution phenological forms of Pedunculate oak (Quercus robur L.) in Donetsk region. Biol Syst. 2016, 8, 272–279.
  2. Utkina, I., Rubtsov, V. Studies of Phenological Forms of Pedunculate Oak. Contemp. Probl. Ecol. 2017, 10, 804-811. https://doi.org /10.1134/S1995425517070101.
  3. Pirko, Y.V., Netsvetov, M., Kalafat, L.O., et al. Genetic features of the phenological forms of Quercus robur (Fagaceae) according to the analysis of the introns polymorphism of β-tubulin genes and microsatellite loci. Ukr Bot J. 2018, 75, 489–500. https://doi.org/10.15407/ukrbotj75.05.489.
  4. Kitin, P. Dynamics of cambial activity in the stem of early- and late-flushing forms of oak (Quercus robur vars. praecox and tardiflora) in the Park of Freedom, Sofia. Nauk za Gorata 1992, 27, 3–13.
  5. Koval, I.M., Kostyashkin, D.C. The influence of climate and recreation on formation of layers of annual wood of early and late forms Quercus robur L. in Kharkiv. Greenbelt. Sci Bull UNFU 2015, 25, 52–58. https://doi.org/10.17221/37/2020-JFS.

2) I suggest that the abstract should be supplemented by the most important finding of the study. 

Reply: We added information regarding forms adaptation and stability of radial increment.

3) The Latin names of the taxa should be in italics throughout the text. 

Reply: Corrected, thanks for the comment.

4) I think that the aim of the paper is not properly formulated. The results of the study should explain the phenomenon, but not simple differences in growth within the one forest. A hypothesis should also be formulated. It is possible to formulate the hypothesis that oaks of a certain race are more adaptive than oaks of another race. 

Reply: We have reworked the introduction, discussed the information of the differences between phenological forms. A hypothesis was put forward and the goal was adjusted.

“However, we assume that the differences in the features of the annual growth of growth rings between phenological forms in the forest-steppe region, at the border of the range, will remain. Oaks of a certain form are more adaptive than oaks of another form, but it is necessary to take into account forest growing conditions The aim of the study was to elucidate the patterns of growth dynamics of English oak trees of both phenological forms at the border of the range in the forest-steppe zone in different types of forest growth conditions.”

5) How were the races determined in the field and how was the health (line 62) of the tree assessed? These are very important methodological considerations that need to be described. 

Reply: The following information has been added to section 2.3:

“Phenological forms were established by phenological observations in the studied forest types during a route survey (the route ran from the Osered River floodplain at the Red Cordon (lat 50.664906 N, log 40.363079 E) to the Chernavsky cordon, the Red Forestry) and during the analysis of sample plots. The accounting of trees began with their entry into the phenophase "bud opening". As a control phenophase, the timing of the beginning of the budding of oak leaves was taken. Observations were carried out at intervals of 5 days. The calculation of the average phenophase (x ) was established using equation 1:       

x ̅=(∑(x⋅n))/Σn                        (1),

where х - observation dates given by the month of registration of this phenophase; n - the number of individuals who entered the phenophase by the last observation;   Σn - the total number of registered individuals in the plot.

The health of the tree assessed in accordance with the current rules of sanitary safety in the forests of the Russian Federation, according to which they are allocated to healthy, weakened, severely weakened, shrinking and dead trees and plantings in general [38].

6) I think that the term 'storey' is completely inappropriate for soil (Table 1). Soil is made up of layers.

Reply: Table 1 "Silvicultural characteristics of the forests." shows the characteristics of the experimental areas under study, in the second column the term "storey" is used to characterize the breed composition of the planting tiers.

7) Line 148. The sentence is unclear and needs to be clarified.

Reply: Was changed to «The dating and measurement of the width of the annual rings (with an accuracy ± 0.05 mm) was carried out using LINTAB 6 ring-width measuring system (Rinntech Com-pany, Heidelberg, Germany) and analyzed the data using the TSAP-Win software pro-fessional programs.»

8) The methodology is not sufficiently clear, especially the statistical part. It is written that ANOVA was used, but it is not stated which one (one-way, two-way, nested)? How were the results of the radial increments assessed? Whether the data sets were normally distributed or non-normally distributed? Only by knowing the answers to these questions is it possible to assess the interpretation of the results.

Reply: We analyzed the obtained data by one-way Regression analysis was used to identify causal relationships between meteorological factors and radial increment.  Empirical data of the radial increment and meteorological factors were entered into the “Aproks” computer program, the closest theoretical coupling equation was calculated and the dependence equation was constructed. The absolute values of the width of the annual increment have an abnormal distribution, since there is an age trend. The relative indices of the annual increment are close to the normal distribution.

Additions have been inserted into the methods.

9) There are unexplained abbreviations in the tables (e.g., Table 3).

Reply: Explanations have been added to the table, some of the columns are called non-abbreviated terms

10) What is indicated next to the mean? Standard error or standard deviation?

Reply: This is «Standard error», was added in Table 3

11, 12) How should the statements in lines 206 to 212 onwards, which deal with tree growth in 1938, 1939, etc., be understood, when all the graphs and tables indicate that growth has been estimated since 1958? And then how to relate the information given in Figure 6, Figure 10, etc. If different periods were taken, then it is necessary to explain all the reasons and provisions in the methodology.

Reply: We have consistently corrected the drawings, added missing periods. Thank you for the important remark.

13) Line 388. What are those specific features?

Reply: In the first version of the discussion, specific features were discussed further in the text. Now the phrasing of the text has been restructured and information has been added on the difference not only between species of the genus Quercus, but also for phenological forms.

14) Line 420. What is P. sylvestris and what is Q. humilis? The rules of nomenclature must be followed, which state that the genus name may only be abbreviated when the abbreviation does not cause ambiguity. 

Reply: Thank you for the comment, it has been corrected.

15, 16) I suggest reorganizing the Conclusions section. Some of the statements should be moved to the discussion section and the sources of information (on race resistance) should be indicated. The authors should edit the content of the paper to avoid unexplained abbreviations (Table 5), contradictions and inaccuracies. 

Reply: We have tried to rework the work in accordance with the comments.

Round 2

Reviewer 1 Report

The revised version of the manuscript is a little better than the previous one. But just a little. Unfortunately, the text is difficult to follow due to numerous errors in the numbering of cited publications, missing some publications in References. There is also a need for thorough language correction first. Many sentences are incomprehensible due to their construction. I will comment after correcting these imperfections if the manuscript gets a chance in the next round of reviews.

Author Response

Dear Reviewer, we thank you for your comments.

Numbering and references was revised. The correction of the language was made. We hope that we were able to correct the shortcomings in the manuscript according to your comments.

Best regards, Konstantin

Reviewer 2 Report

Most of the comments made in the previous review have been adequately taken into account by the authors, who have corrected the article or substantiated their views.  The quality of the article has improved significantly. Nevertheless, there are still a few issues that the authors should correct.

1. The whole introduction is now composed of two long paragraphs. The text needs to be divided into logically self-contained paragraphs, which would make the text easier to read and understand. 

2. I suggest that the authors critically read the text of the article and optimise it, avoiding unnecessary sentences that do not mean anything (e.g., Although annual precipitation is fairly significant, it is very uneven; line 14). What do the authors want to say with this sentence? When precipitation is significant, when it is insignificant. In this case, the adverbial clause only confuses the reader. There are a lot of similar sentences in the text.

3. The authors should be very careful and precise in their use of statistical terms. If something is said to be significantly different, the significance of the difference should be stated (e.g., lines 280, 371, 445 etc.). I could not find a single reference in the paper indicating how the values analysed differ significantly between year groups, periods or phenological forms. Why is there no indication of the significance of the difference (H, U, t or other values depending on the statistical method used to determine the difference)?

4. The text is full of technical errors which in some cases are even confusing. For example, the year ranges are written in three different ways (e.g., 1997-1998; 1997–1998; 1997 – 1998). 

Additional note:

In their replies to the comments made in the first review, the authors have referred to a number of articles in support of their choice of terminology. I agree that it is important to use the terminology adopted by other authors, but it is also necessary to check the meaning of the terms against specialised or reputable English dictionaries. These are very clear on the meaning of many terms and do not give rise to further discussion. The use of English terms in local publications, even if they are published in English, needs to be particularly critical. 

Author Response

Dear Reviewer, we thank you for your careful reading of the manuscript and your comments. The necessary editing of the manuscript was carried out and we hope that we managed to eliminate all the shortcomings according to your comments.

Answers to your comments in the attached file.

Best regards, Konstantin
